# A Review of Biopolymers’ Utility as Emulsion Stabilizers

**DOI:** 10.3390/polym14010127

**Published:** 2021-12-30

**Authors:** Nirmala Tamang, Pooja Shrestha, Binita Khadka, Monohar Hossain Mondal, Bidyut Saha, Ajaya Bhattarai

**Affiliations:** 1Department of Chemistry, Mahendra Morang Adarsh Multiple Campus (M.M.A.M.C.), Tribhuvan University, Biratnagar 56613, Nepal; nt891840@gmail.com; 2Central Department of Biotechnology, Tribhuvan University, Kirtipur 44618, Nepal; shrestha.pooja2012@gmail.com (P.S.); khadkabinita1000@gmail.com (B.K.); 3Chemical Sciences Laboratory, Government General Degree College, Singur 712409, India; 4Homogeneous Catalysis Laboratory, Department of Chemistry, The University of Burdwan, Burdwan 713104, India

**Keywords:** emulsion stabilizer, nanoemulsion, emulsion technology, biopolymer

## Abstract

Polysaccharides, polynucleotides, and polypeptides are basic natural polymers. They have various applications based on their properties. This review mostly discusses the application of natural polymers as emulsion stabilizers. Natural emulsion stabilizers are polymers of amino acid, nucleic acid, carbohydrate, etc., which are derived from microorganisms, bacteria, and other organic materials. Plant and animal proteins are basic sources of natural emulsion stabilizers. Pea protein-maltodextrin and lentil protein feature entrapment capacity up to 88%, (1–10% concentrated), zein proteins feature 74–89% entrapment efficiency, soy proteins in various concentrations increase dissolution, retention, and stability to the emulsion and whey proteins, egg proteins, and proteins from all other animals are applicable in membrane formation and encapsulation to stabilize emulsion/nanoemulsion. In pharmaceutical industries, phospholipids, phosphatidyl choline (PC), phosphatidyl ethanol-amine (PE), and phosphatidyl glycerol (PG)-based stabilizers are very effective as emulsion stabilizers. Lecithin (a combination of phospholipids) is used in the cosmetics and food industries. Various factors such as temperature, pH, droplets size, etc. destabilize the emulsion. Therefore, the emulsion stabilizers are used to stabilize, preserve and safely deliver the formulated drugs, also as a preservative in food and stabilizer in cosmetic products. Natural emulsion stabilizers offer great advantages because they are naturally degradable, ecologically effective, non-toxic, easily available in nature, non-carcinogenic, and not harmful to health.

## 1. Introduction

“Biopolymers are natural sources polymers which are obtained either synthesized chemically from a biological material or biosynthesised entirely by living organisms” [1], such as proteins (amino acid polymers), genetic material (nucleic acid polymers), glycoforms (carbohydrate polymers and glycosylated molecules), metabolites, and other structural molecules. These polymers can be incorporated in two different ways, depending on whether they are surface-active (most proteins) or surface-inactive (most polysaccharides) [2]. They perform several functions such as coating, packaging, and many other mechanical functions, according to their properties. They are biodegradable and do not accumulate like many other synthetic polymers, such as waste plastics [3].

The emulsification properties of polymers are based on their monomer units and are used in a variety of ways depending on their basic features in different fields, such as food, pharmaceuticals, cosmetics, pesticides, fertilizers, etc. The overall biopolymer types are listed in Figure 1.

These polymers are obtained from organic materials such as microorganisms, organic monomers, and biomass. The biomass obtained from biopolymers includes polymers of amino acids, polymers of carbohydrates, and glycosylated molecules. The main sources of polymers of amino acids are animal proteins, such as whey proteins; plant proteins such as soy proteins, pea proteins, etc.; and polysaccharides derived from cellulose, starch, alginate, chitosan. Biological cellulose and polyhydroxyalkanoates (PHAs) are derived from organic materials and bacteria [5]. The different types of biopolymers obtained from natural sources are listed in Figure 2.

### Application of Biopolymers

When used individually or in combination, biopolymers are excellent stabilizers in tests. Polymerized composites are favored in the food production technology due to their durability, less-poisonous nature, less immunogenic character, bio-adaptability, fine chemical reactivity, comparative cost-effectiveness [6,7], stability, nutritional benefits, and biodegradability [8]. They have also been found safe to use. It has long been common practice around the world to use many substances as flavors, preservatives, and emulsifiers in foods to enhance their color, taste, or texture. The demand for natural food products is driving the growth of these biopolymer alternatives to synthetic stabilizers for emulsions. It is astonishing to witness the success with which biopolymers have been mechanically implemented in such diverse industries as nutrition, microbiological, medicinal, cosmetics technology [9], and mining [10], with proteins and polysaccharides being the most frequently used biopolymers, often in forms of colloidal diffusion synthesis such as froths or emulsions [9]. Research suggests that sugar metabolism and metabolic syndrome can play a significant role in treating inflammatory bowel disease and obesity. Bio-compatible polymers are more eco-friendlier and degraded by using enzymes and hydrolysis. Table 1 is the tabulated form of biopolymers and biopolymer degradation and their sources.

One of the most important applications of biopolymers is the synthesis of biodegradable plastics. Currently, plastic is one of the most commonly used materials. Most chemically synthetic plastics are non-degradable. This could create great problems in the near-future. Therefore, by using biodegradable plastics, we can replace non-degradable synthetic plastic. About 0.5% of bio-compatible polymers, such as Polylactic acid (PLA), Polyhydroxyalkanoate (PHA), Polyhydroxybutyrate-valerate (PHBV), and Cellulose, are used as plastics [12].

When biopolymers are used as emulsion stabilizers, they display a variety of modes of action. Additionally, they act as emulsifiers (polypeptides), viscosifiers (polysaccharides), and weighting agents (polysaccharides and polypeptides) while decreasing coalescence by coating individual droplets. Chemical, enzymatic, or thermal treatments can be used to covalently complex biopolymers. As a result, the final complexes are generally more robust and soluble. According to studies, biopolymer complexes exhibit higher temperature stability, pH stability, and ionic strength stabilization.

In terms of food structure and stability, biopolymers have made significant contributions [13]. Their techniques include progressing the viscosity of the regular phase to minimize globule movement [14], producing a soft covering around each oil globule to minimize coalescence, and expanding the oil globule density to reduce creaming. In addition to increasing product shelf life, all these mechanisms increase emulsion stability. Essentially, proteins are surface-active, enabling them to bind and emulsify aqueous polysaccharides, which alter the viscosity of the aqueous phase to slow the migration of droplets. With polysaccharides and proteins, lower concentrations of the polymer result in better-stabilizing properties. Similarity and sustainability can be achieved by optimizing the concentration of polymers, understanding how they are complex, and understanding system nature. Biopolymers are not only able to increase the stability of emulsions, but also to improve their nutritional value, as well as affecting their shelf life, texture, and oral consistency. A wide variety of uses can be found for biopolymers, such as nutrient delivery in food products and bioactive compounds.

## 2. Emulsion and Emulsion Stabilizer

### 2.1. Emulsion and Emulsification

In an emulsion, immiscible droplets are dispersed between two liquid phases. An example is the dispersion of water in oil or the dispersion of oil in water when it is stabilized by a suitable surfactant [15]. The substances using stabilizer agents in emulsion and nanoemulsion are called emulsion stabilizers; the general number of biopolymers used as monolayer stabilizers as well as multilayered stabilizers is presented in Figure 3. Nanoemulsion and emulsion are two different substances, with significant differences between them. The main differences between emulsion and nanoemulsion are listed in Table 2.

### 2.2. Stabilization and Destabilization of Nanoemulsion

Although nanoemulsions feature small droplet sizes, they exhibit long-term stability due to their ability to withstand destabilization processes such as creaming, sedimentation, and coalescence, as shown in Figure 4. Nanoemulsions have been used to solubilize and preserve drugs against unpleasant environmental factors in the parenteral form, such as oxidation, pH, and hydrolysis [19], to target special fixed organs by exploiting the increased absorptivity and reservation effect [20] and to evade the reticular endothelial system [21]. Nanoemulsion droplets are large enough to saturate highly hydrophobic drugs and increase their dissolution, resulting in an anticipated increase in their systemic bioavailability [22]. As nanoemulsions partition and diffuse from the oil to the surface-active layer and then into the hydrolyzed stage [23], they offer the possibility of obtaining sustained/controlled release devices. During nanoprecipitation, the drug’s surface area is greatly increased, which ultimately accelerates its dissolution. The Noyes–Whitney equation is used to calculate the dissolution rate. Sedimentation occurs when particles in suspension are trapped in a medium and then settle into it. Therefore, preventing such aggregation may stabilize an emulsion. The suspension is unstable due to the presence of flock-like colloids, known as flocculation. In flocculated systems, the Van der Waal forces are stronger than the repellent forces, which is why droplets tend to stick together. To prevent flocculation, a solution must overcome the attraction between droplets. Electrical double layers can be formed with anionic surfactant to create repelling forces. Emulsions become unstable when two or more droplets coalesce, resulting in coalescence. To reduce coalescence, polysaccharides and polypeptides act as weighting agents.

Additional elements such as the ability to convert direct para-cellular/transcellular transfer [25,26] extend gastric retention because of mucosal entanglement [27], as well as helping in nanoemulsion-mediated bioavailability improvement. Nanoemulsions have been shown to absorb directly into the lymphatic system, reducing the likelihood of first-pass metabolism and improving the bioavailability of drugs subject to hepatic transformation to a great extent [28]. Across a variety of sectors, including food technology, pharmaceuticals, and agriculture, biopolymers are used to form nanoemulsions to an increasing degree. Colors, flavors, lipids, preservatives, vitamins, and nutraceuticals are among the hydrophobic functional ingredients that food and beverage producers must encapsulate in their products. By encapsulating these functional ingredients, they may be more easily handled, water-dispersible, and chemically stable. Biologically active substances such as vitamin A, D, E, F, lutein, cumin, and coenzyme Q10 can be encapsulated, protected, and delivered effectively by nanoemulsion-based delivery methods [29].

### 2.3. Synthesis and Application of Nanoemulsion

Nanoemulsions can be made and used in several dosage forms, such as fluids [30], pastes [31], fogs [32], gels [33,34], fine particles of liquid and solid in the air [35,36] and can be equally applied by changing routes such as topical [37], oral [38], intravenous [39], intranasal, pulmonary and ocular [40]. In addition to the cosmetic industry [41] and pesticide industry [42], they also have been used as aqueous mediums for organic deliverables due to their superior solubilization capabilities. We compared the nanoemulsions with their droplet size obtained by different techniques used from various sources, as shown in Table 3.

## 3. Polysaccharides Chemical Structure and Their Properties

### 3.1. Synthesis of Polysaccharides Emulsion

Monosaccharide units, also known as glycosyl units, are used in polysaccharides to create larger molecules [43]. A polysaccharide’s degree of polymerization is identified by the number of monosaccharide units in it; the degree of polymerization (DP) ranges from 100 to more than 10,000, with the majority falling between 200 and 3000. Homoglycans and heteroglycans are distinct in that they are composed of different sugar monomers. The former, for example, are formed by monomers of the same sugar in starch amylose, while the latter are made up of different monomers. Heteroglycans such as align and guar gum are also found in locust bean gum [44].

To lower its interfacial tension, alkyl polysaccharides are produced against hydrocarbons phases and the use of polysaccharides with ions in personal care products. In shampoo, shower bath, and soap formulations, ionic surfactants produce more foam than non-ionic sugar surfactants.

### 3.2. Glycosyl with Polysaccharides

The presence of glycosyl units, which contain three hydroxyl groups, makes polysaccharides very hydrating, since water molecules are highly attracted to them. Polysaccharides are hydrated more easily because they can form bonds with water, called hydrogen bonds [45,46].

Furthermore, glycosidic oxygen atoms can form hydrogen bonds with water when the oxygen atom is added to the ring structure. The functional properties of foods such as texture can be modified and controlled by carbohydrates with lower molecular masses by controlling the mobility of water in the food system [47]. It is important to understand that water whose structure has been sufficiently modified by the polymer so that it does not freeze is sometimes called polymer or polymerizing water. The water that is naturally hydrogen-bonded to polysaccharide molecules is called non-freezable water. A chemical scale does not show the molar bonding of these molecules in this water. Regardless of their slowed motion, they are free to exchange with other water molecules and can do so rapidly. There is little water contained in gels and fresh tissue foods other than the water that is essential for hydration. Water is entrapped in gels and tissues in t capillaries and holes of different sizes above the hydration water level. Rather than cryoprotectants, polysaccharides act as cryostabilizers. Their large size and high atomic weight mean that they do not significantly affect water’s osmolality or freezing point, and their colligative properties do not cause the molecules to behave in these ways [48]. When polysaccharide solutions are frozen, crystalline water (ice) and glass, containing perhaps 70% polysaccharide molecules and 30% non-freezable water, are created. Non-freezable water is such a complex mixture of molecules that it is part of a fluid whose molecules feature very little mobility due to its very high viscosity, which is only possible in carbohydrates of lower molecular weight. Others provide cryostabilization by adsorbing to nuclei or active crystal growth sites to limit crystallization by freezing a freeze-concentrated matrix.

Polysaccharides differ in their properties based on their molar mass, electrical double layers, hydro-basicity, polarity, and branching degree. Glycoproteins or glycolipids covalently bound to polysaccharides improve their emulsification performance.

## 4. Food Protein and Food Protein Emulsions

### 4.1. Animal Protein and Plant Protein

Proteins, which are widely used in food, are most often extracted from animal sources to be used in pharmaceutical applications, such as microencapsulation. Despite various appealing properties of animal protein for microencapsulation, such as smooth/high solubility, lower molecular mass, flexibility (and, hence, greater stabilizing), entrapment, and oxidation-resistant properties compared to plant-based proteins, plant proteins command more attention from consumers as animal products raise the question to food safety due to some health-related controversies related to animal products. One example is the risk of Bovine Spongiform Encephalopathy (BSE) (also called transmissible spongiform encephalopathies (TSE)), a fatal neurodegenerative disease affecting humans and animals, caused by the abnormal formation of a cell protein called prion protein (PrP)). In comparison with animal proteins, plant proteins are more economical and readily available. There is also the possibility of allergy to plant proteins. Therefore, reducing allergenicity requires careful selection of the plant protein (for example, pulse proteins).

A microencapsulation experiment on sweet orange oil [49] utilized Soy Protein Isolate-Gum Arabic (SPI-GA) coacervates to find the different effective factors, such as ionic character, the ratio of SPI/GA, pH, basic elemental load, and introducing sucrose and maltodextrin in the composite and effectiveness of microencapsulation. Eventually, in 4.0, 0 mol/l NaCl pH with a 1:1 ratio of SPI to GA and a 10% loading core materials were found to produce the maximum coacervate yield and encapsulation of basic microelements. Moreover, microencapsulation yields increased dramatically when sucrose was coupled with SPI (sucrose: SPI ratio 1:1). According to previous research [50], the spray-drying of soybean oil produced SPI microencapsulation was a factor influencing retention ability, re-distribution or dissolution properties, and stability in store. Core–wall ratios of 1:1 or higher negatively affect redispersion characteristics, while a 1:1 wall to core ratio exerts a higher positive effect on them.

### 4.2. Effectiveness of Plant Protein

The non-modified status of pulse proteins has, along with their reduced risk for allergens, made them a popular alternative to soy because they are considered a superior replacement. By using a complex coacervation process, Ducel et al. examined pea globulin used as a wall material in the microencapsulation of model oil. They also studied and compared a cereal protein (alpha-gliadin) and a leguminous protein (pea globulin). The effect of pH and the protein–anionic chemical ratio value were the main topics of investigation for them [51]. In a similar study by Gharsallaoui et al. in pea protein microcapsules with Miglyol 812 N = 5%, pea protein = 0.25%, and maltodextrin = 11% in pH = 2.4 with the addition of spray-dryer, then recalculated at pH = 2.4, as a model oil, stability-to-droplet aggregation was enhanced when the pectin coating was applied after drying. In addition to maintaining oil droplets in suspension, pectin also increased steric repulsion [52].

### 4.3. Pea Protein Entrapment Efficiency

Karaca et al. found that spritz-dried pea protein-maltodextrin and lentil protein capsules featured maximum entrapment capacities of 88% and 86%, respectively, and released 37% and an additional 47% of the closed flaxseed oil after 2 h and 3 h, respectively [53]. From using freeze-drying, 35.5% maltodextrin-DE9 and 10.5% oil was found to be an optimal wall formulation that accorded good entrapment ability about 83%, the smallest surface oil of about 3%, and a suitable average globule width of about ~3 mm [53]. Furthermore, as the emulsion oil content expanded, the diameter of the oil globule and the surface area of the oil content increased, whereas entrapment efficiency decreased.

### 4.4. Other Plant Protein and Entrapment Efficiency

Other plant proteins rarely employed as encapsulating agents are cereal grain proteins. Researchers have studied how to enhance the properties of these proteins. Using a highly concentrated zein protein extract from corn gluten, a spray drying process was used to encapsulate tomato oleoresin. Zein concentrations of 1 to 10% (*w/v*) increased entrapment efficiency from 74% to 89%, but no further increase in entrapment efficiency was observed at 14% [54]. Wang et al. encapsulated fish oil using barley protein taken as a microfluidizer in proteins = 15% and using oil protein ratio = 1:1, followed by spritz dryness processing (at 150 °C), with loading efficiencies of 50% [55]. Jiang et al. changed the initial soy proteins structure by acid pre-treating in pH 1.5 to 3.5 and in alkaline solutions with a pH = 10–12 for many repetitions (0–4 h); the results showed an increase in surface hydrophobicity in the form of a protein-adjusted, liquefied droplet-type configuration, and beneficial changes to its emulsifying characters [56]. Augustin et al. identified that increased temperature-time reactions were necessary to increase the stability of fish oil microcapsules after emulsification. They also observed that heat treatment increases entrapment efficiencies [57].

### 4.5. Modification of Protein into Functional Components

There are many ways to modify plant proteins chemically or enzymatically. The controlled deamidation and glycosylation of rice endosperm protein emulsion were achieved by Paraman et al. [58]. A denaturation process during protein synthesis could contribute to increasing protein hydration. A methanol-alkali deamidation improvement of rice endosperm protein’s emulsifying properties was found to be the most effective. Wong et al. found that many locations of conjugation and dextran were dependent on the dextran size in wheat protein-dextran Maillard conjugates prepared using the deamidation method [59]. In comparison with the adsorption of protein alone, the complexes were shown to create a deep interfacial, coverage providing greater steric stabilization. Glycation enhanced the emulsifying properties of kidney bean vicilins in the presence of glucose described by tertiary conformation unfolding and rearrangement, and increased quaternary flexibility [60]. Polysaccharides have been found to enhance emulsion stability when used in a mixture with proteins in emulsion [61] by raising the durability of the interfacial thin layer dividing the globules and reducing the droplets’ movement rate, altering the viscosity of the regular phase. Maltodextrins are favored as subsidiary wall materials or additional elements of microencapsulation to upgrade the drying capacities of microcapsules because they possess fine solvability and lower viscousness in maximum solid content [62].

## 5. The Stability Factors of Proteins Nanoemulsions

Many factors contribute to emulsion stability, such as fluctuating temperature and pH, storing age, ionic strength, and processing technology [63], where temperature and time are the two main determinants. Temperature and time significantly affect the increase in the size of particles, the retention of β-carotene, and the rate of potential decrease with storage temperature as well as time [64]. In addition to droplet collisions in storage, the liquid phase separation rate of nanoemulsions is affected by Brownian motion at higher temperatures. This behavior leads to an increase in the size of the particle due to mass transfer kinetics between the water and oil phases [65,66]. A qualitative study showed that protein emulsifiers inhibit lipid oxidation more effectively than small-molecule surfactants [67]. The food industry will be able to make better use of that antioxidant by utilizing carotenoids, which include β-carotene. In an experiment conducted by [63], a glucamine-based trisiloxane surfactant was obtained through the green synthesis technique. Studying various physicochemical characteristics of the compound, including its surface functioning, accumulation, and wetting properties, HAG (4-S-Glutathionyl-5-Pentyl-Tetrahydro-Furan-2-Ol) was reported to feature a relatively lower surface tension (γ = 19.04 mN/m) and to interact readily with surfaces. It has also been demonstrated that HAG reduces surface tension with remarkable efficiency. A highly assemblable microdevice can also be used for encapsulating drugs and delivering them, as well as a microreactor, as evidenced by TEM and dye encapsulation experiments. Additionally, it could be used as an adjuvant, a cleaning agent, a coating, or a home care product. In total, 99% of HAG can be biodegraded within a week using primary biodegradation experiments.

### 5.1. Encapsulation and Encapsulation Efficiency

Various delivery systems for multifunctional drugs, e.g., nanoparticles [68], liposomes [69,70], nanogels [71], nano-capsules [72], and copolymer micelles [73,74,75] have been extensively explored in doxorubicin (a chemotherapy drug called anthracycline, which blocks topoisomerase 2, which cancer cells need to divide and grow) delivery systems to increase antitumor efficiency for a few decades. Polymeric micelles have been attracted to enhancing attention in the form of beneficial nano-carriers for transporting antitumor drugs owing to their superior characteristics, such as self-assembling into micelles in solution, greater consistency besides the reduction of blood density, extended reservation, and superior tumor assembly [76,77,78,79]. A similar study on assorted micelles of Dox@FA-BSP-SA/TPGS regulated under the supervision of Liu et al. [80] reported that the cytotoxicity environment and anti-tumor effectiveness in vivo outcome of Dox@FA-BSPSA/TPGS micelles was ranked than that of doxorubicin-free and Dox@FA-BSP-SA individual micelles, suggesting that it is a promising candidate as a drug delivery carrier for cancer chemotherapy. FA-BSP-SA/TPGS combined micelles presented higher biocompatibility, with a moderate elemental size of 147.3 nm, a load capacity (LC) = 14.4%, and a encapsulation efficiency (EE) = 91.9% for doxorubicin, with a weight ratio of 3:1.

### 5.2. Emulsifying Properties of Proteins

The protein, or polysaccharide, plays an important role as an encapsulating agent. Such encapsulating agents protect sensitive elements from harmful environmental agents such as oxygen, temperature, pH, moisture, etc., as well as preventing unpleasant smells and flavors, helping to uniformly disperse the active ingredients, and simplifying the handling of the active ingredients [81]. The oils containing *n*-3 polyunsaturated fatty acids, such as linolenic acid, docosahexaenoic acid, and eicosapentaenoic acid, have been entrapped; these oils receive the most attention because they feature healthy characteristics in infant growth, minimizing the possibility of heart and blood vessel-related diseases, and preventing swelling [82,83,84]. When plant protein ingredients are used, they feature less solubility, reduced emulsifying abilities, and decreased reactivity to crosslinking factors relative to proteins formulated from animals, such as whey and casein. Some protein-emulsifying properties are as follows:(a)Surface hydrophobicity: The percentage of hydrophobic proteins exposed on the surface of proteins measures how much of the protein can adsorb to the oil phase. The presence of hydrophobic sites buried inside proteins can be revealed by partial denaturation, which can increase their emulsifying ability [85].(b)The flexibility of proteins: It is a self-rearrangement property of proteins, when it is adsorbed at the oil–water (O/W) interface, most of the hydrophilic mass favors the watery parts and the hydrophobic mass favors the oily parts by reducing the attractive force between two liquids [86]. According to the composition of protein, hydrophilic loops of amino acids may enlarge away from the O/W interface in the form of waterish parts, slowing the reaction [86].(c)The dimension of the protein molecules may affect their movement on the O/W interface emulsification process, and the film formation capacities of the protein. Luyten et al. (2004) [87] reported that smaller proteins are more effective for diffusion at the interface than larger proteins [85].(d)When encapsulating, a high solubility of proteins is preferable in order to allow higher movement in the O/W interface and higher continuous phase viscosities [85].(e)The factor influences in the protein solubility are the pH of the solvent, the ionic character, and the attractive or repulsive forces between closer globules showing emulsion instability or stability. When the solvent pH is not near the isoelectric point of proteins or when ionic conditions are low, charge repulsion can enhance emulsion stability [88].

In a study by Liu et al., four major macromolecular proteins with emulsifying abilities were studied, which included peanut protein isolate, whey protein isolate, rice bran protein isolate, and soy protein isolate [89].

### 5.3. Protein from Rice Bran as an Emulsifier

Rice bran protein (RBP) features high surface activity and good hydrophilicity [90]. Therefore, it is suitable for emulsifiers to stabilize nanoemulsions. Based on one experiment, rice bran protein-based nanoemulsions feature the lowest globule dimension and are highly stable at RBP = 3% and pH = 9.0. When quercetin was added to nanoemulsions, the resulting nanoparticles were smaller and more organic. In an alkaline medium with low concentrated salt ion, the rice bran protein-based nanoemulsion was stable. The use of RBP NEs resulted in a 12.70 ± 0.12% increase in quercetin bioavailability after in vitro digestion and cell piercing observation. In addition to reducing quercetin’s toxicity to cells, nanoemulsion-encapsulated quercetin increased the degree of penetration into cells, reaching 4.93 ± 0.01 × 10^−6^ cm/s. The RBP NEs and QE-RBP nanoemulsions both feature a 14 day preservation period. RBP NEs can be used to carry biologically active molecules, according to this study [90].

## 6. Phospholipids Nanoemulsion Stabilizer

### 6.1. Phospholipids

Phospholipids (PLs) are ubiquitous and play a significant structural role in biological membranes [91]. Natural phospholipid purification and processing features less environmental impact and lower energy costs than synthetic methods. Natural phospholipids are considered eco-friendlier because they are manufactured using environmentally friendly methods and made using renewable raw materials [92]. Compared to their synthetic counterparts, they are cost-effective and provide health benefits.

### 6.2. Type of Phospholipids and Application

Phospholipids function well as excipients in pharmaceutical formulations. They can take a wide variety of forms, such as fat emulsions, a combination of micelles, suspensions, and preparations of liposome using any administration route. As natural, effective substitutes for synthetic emulsification agents, such as polysorbates, polyoxyethylene, castor oil derivatives, sucrose, fatty acid, esters, etc., are effective alternatives. These molecules are used for emulsification, wetting, solubilization, and liposome formation because of their amphiphilic nature i.e., one group is characterized as a polar head and another is characterized as a lipophilic tail.

Phosphatidylcholine (PC), phosphatidylethanolamine (PE), and phosphatidylglycerol (PG) are the typical phospholipids found in membranes. The lecithin derived from vegetable oil is a mixture of PC, PE, phosphatidylserine, and phosphatidylinositol, as well as fatty acids, carbohydrates, and triglycerides. Grades of lecithin containing more than 80% PC are also called PC; the grade that contains less than 80% PC is known as lecithin. Currently, “lecithin” is used as the commercial name for the combination of phospholipid specifically used in the cosmetic and food industries [93]. All cell membranes are made up of phospholipids, which function structurally and as functional molecules. The function of phospholipids and other membrane components in signal transduction can be explained by the interaction of phospholipids and other membrane components. Beyond the useful characteristics of phospholipids in the outer membranes of the cells, phospholipids also perform well at metabolic functioning in bile to dissolve fatty components and cholesterol as monoacyl-phospholipids, which humans consume through meals and lipophilic drugs [94]; as lipoprotein substances for the movement of fats between liver and gut; as sources of acetylcholine; and also as sources of energy and essential fatty acids [95]. Moreover, in lung surfactant, a specific phospholipid, dipalmitoylphosphatidylcholine, occurs [95]. Phosphatidylserine is a particle of the lipid calcium phosphate conjugates for accumulation in the bone-forming process [96], control of apoptosis [97], and blood clotting [98].

### 6.3. Natural Phospholipids

Natural phospholipid recipients refer to phospholipids that can be obtained from natural sources, such as rapeseed, soybean, sunflower seed, etc. Unsaturated phospholipids are changed into saturated phospholipids by hydrolysis or enzyme treatment methods, such as the conversion of partial fatty acids into the polar head group. Naturally occurring saturated phospholipids feature their own natural identities. By comparison, synthetic phospholipids are phospholipids that are synthesized through the addition of specific molecules, such as fatty acids, through a tailor-made chemical synthesis process, among others. In addition, PLN-encapsulated phospholipid complexes are more stable and effective in vitro against cancer [99]. The phyto-phospholipid complex is also used to improve the bioavailability of poorly absorbed phytopharmaceuticals through oral administration [100]. Phospholipids can be used in the treatment of lung cancer due to the way phospholipid complexes can encapsulate, uptake, and act as anti-tumorigenics [101].

## 7. Advantages and Limitations of Biopolymers over Synthetic Polymers

### 7.1. Challenges of Synthetic Polymers

As one of the first synthetic polymers ever produced, plastic is a prominent competitor of biopolymers. The fossil fuel industry provides the majority of the feedstocks for plastic manufacture. With time, the usage of plastics has become more common; however, the availability of petroleum and fossil fuel is reducing. In addition, petroleum-based plastics exert negative environmental effects since they are made from carbon, which has been trapped in the earth for millions of years. Therefore, releasing these carbons through incineration or other methods leads to a greater amount of greenhouse gas emissions in the atmosphere [102,103]. The lifespan of the synthetic polymers, which could have been a useful property, has also become a drawback due to its overuse and the inability to manage and recycle it effectively, since it manages to remain in the environment for a longer time in various forms, such as polyethylene terephthalate (PET), high-density polyethylene (HDPE), plastics for packaging, etc.; the recycling rates are 29.1%, 29.3%, and 8.7%, respectively as of 2018 [104]. These numbers are insignificant compared to the threat posed by these products to the environment and the earth’s ecosystem, aggravating their devastating consequences. Therefore, the quest for a better alternative to synthetic polymers, for which the biopolymers can be promising candidates, is more urgent than ever.

Environmental pollution has become a major problem in the past decade, and the public has become aware of it, which has led to an increase in the demand for environmentally friendly products that utilize biopolymers, such as lipids, polysaccharides, and proteins, among others, that are inexpensive renewable raw materials that can be considered as an alternative to petroleum-based, non-biodegradable plastic products [105,106]. They can be derived from a wide variety of feedstocks, including agricultural products such as corn or soybeans, and alternative sources such as algae or food waste.

In addition to traditional sources, such as corn and soybeans, biopolymers can also be derived from non-traditional sources, such as algae and food waste [107,108,109]. Mostly, biopolymers are eco-friendly and inexpensive; their waste processing is an alternative to fossil resource exhaustion (limiting the usage of fossil resources), and lower global warming potential/burden; they are also biodegradable, compostable, and sustainable, with high recyclability and less eco-toxicity [110]. Especially in fields such as medicine, agriculture, engineering, and textiles, where biodegradability and compatibility are critical, these unique features make them different from synthetic polymers.

However, being sourced from renewable sources does not ensure biopolymers’ favorable performance over petroleum-based polymers [111]; hence, sustainability studies, such as life cycle assessments (LCAs), are performed to compare and improve the environmental impacts of biopolymers [112]. Furthermore, from an economic perspective, the cost of biopolymer products should be borne in mind since their future heavily depends on their ability to compete with synthetic polymers in terms of price, despite their valuable properties. Because most biopolymers are expensive, and petroleum-based polymers are less expensive, industries have adopted them without regard for environmental considerations [113,114]. Thus, if this one economic challenge is overcome, the use of biopolymer might accelerate tremendously, exerting a positive impact on the environment and overall wellbeing in a short period. This will help to reduce degrading, allowing the time to reverse the causes of serious environmental catastrophe, such as global warming.

### 7.2. Biopolymers

Latex and cellulose have been used as biopolymers since 1850 for the manufacture of rubber and plastics. Polyhydroxyalkanoate (PHA), which is used in a variety of industries from, medicine to agriculture, and polylactic acid (PLA), which is used to make PLA-based polymers, are made from renewable feedstocks, such as glucose, sucrose, and vegetable oil, etc., through fermentation with lactic acid [115,116]. It is possible to biodegrade and compost PLA-based plastics, providing a wider range of disposal options. To improve the heat resistance or durability of the material, PLA can also be mixed with polymers and fibers derived from petroleum [115,116,117].

The recent discovery of thermoplastic starch (TPS) as a viable alternative to the synthetic polymers often used in packaging might mark a turning point in the history of sustainable materials. TPS is integrated into composites with synthetic polymers to design market-relevant materials that can be utilized in the production of films, rigid materials such as plates and cutlery, packaging, and foams; it also offers the possibility of compostability or biodegradability, depending on constituents [118,119].

Among natural polymers, *n*-acetylanhydroglucosamine and anhydroglucosamine are both found together in chitin as supporting and protecting elements in animal exoskeletons, as well as the exoskeletons of fungi and yeast. A protein and calcium carbonate matrix surrounds chitin molecules, which feature widths of 3–50 nm. Chitosan produced by the deacetylation of chitin is common in marine environments and is the biopolymer of choice for biodegradable polymer films because it dissolves in acidic solutions before assimilating. As a result, chitosan receives a significant amount of attention for its biological and therapeutic activity, as well as its antimicrobial and antitumor effects [120,121].

### 7.3. Comparison in Biodegradability

Conventional plastics are non-biodegradable. By contrast, polymer matrixes and fiber reinforcements are sourced from natural sources, such as flax or hemp. Carbon dioxide and water are the by-products of microorganisms consuming these materials. In this case, biopolymers can be collected and then composted along with bio-waste.

Environmentally friendly by-products, such as carbon dioxide and water, are left by this process [122], allowing the by-products to enter the carbon cycle and the water cycle, respectively. Biological polymers can also be obtained through the microbial fermentation process by utilizing microbial biopolymer feedstock. The resultant products are naturally degradable, environmentally friendly, and suitable substitutes for synthetic plastics. PHA is also accumulated by bacteria as intracellular carbon reserves during periods of nutrient deficiencies. Moreover, microbially produced polyesters are also biopolymers that feature water resistance and thermoplastic properties similar to synthetic plastics. According to one study, when the carbon-to-nitrogen ratio (C:N) in a chemical wastewater treatment system is increased, the specific polymer yield (or PHA production) will also increase [123].

PHA and PLA, although synthetic polymers, are wholly biodegradable and, hence, are considered biopolymers despite not being found in their natural form.

Various biopolymers and synthetic polymers feature different degradability properties, as shown in Table 4.

It should be noted, however, that most biopolymers do not decompose spontaneously in nature, but only when given specific conditions, such as composting. In addition, not all biopolymers are compostable because there are two definitions: either they are polymers from renewable sources (but not necessarily biodegradable) or they are polymers from fossil resources (but not necessarily compostable) [110].

The use of cost-benefit analysis based on the flash pyrolysis of waste biopolymers by Kuppens et al. [132] in certain biopolymers, such as polyhydroxybutyrate (PHB), poly(butylene adipate-*co*-terephthalate), potato starch, PLA, cornstarch, and solanyl (starch-based resin), can offer economic benefits. In the short term, flash copyrolysis of biomass and waste biopolymers was considered the only commercial solution for integrating biopolymers into the plastics industry. 

### 7.4. Comparative Life Cycle Assessment

In the life cycle of a product, the life cycle analysis (LCA) reveals and measures the product’s environmental impact. Biopolymers, for example, can measure and compare their environmental credentials. As a result of the system, a model is created that includes all the process inputs and outputs that occur during the creation of a product. The product is divided into four phases: production, distribution, use, and final disposal or recycling. Life Cycle Assessment methodologies provide appropriate frameworks for assessing sustainability. LCA studies tend to use “cradle-to-grave” and “cradle-to-gate.” systems. A cradle-to-gate LCA considers all the steps involved in obtaining raw materials, converting them, and delivering the final product to the receiving location. These assessments are usually conducted by material producers. Furthermore, cradle-to-grave systems incorporate additional assessment criteria, including all the phases of the cradle-to-grave process, such as the usage and disposal phases, as well as all the stages of the cradle-to-grave process. The impact categories often taken into account in the LCAs of biopolymers are global warming, acidification, eutrophication, ozone layer depletion, smog, and fossil fuel depletion.

Compared to polypropylene (PP), polyethylene (PE), and polystyrene (PS), PHB production results demonstrated a decrease in greenhouse gas emissions. Polymer cradle-to-gate LCA showed impacts in all the impact categories: abiotic depletion, global warming, ozone layer depletion, human toxicity, freshwater aquatic ecotoxicity, marine aquatic ecotoxicity, terrestrial ecotoxicity, photochemical oxidation, acidification, and eutrophication [133,134,135,136,137]. An LCA assessment demonstrated that the PHB production process resulted in the emission of greenhouse gases in the lower amount in comparison to its counterparts (i.e., PP, PE, PS, and PET) [9]. Overall, PHB was found to be more suitable than polypropylene (PP) production in all LCA categories, whether in terms of CO_2_ equivalent production (80% less), ozone layer depletion (50 times lower), terrestrial toxicity (10 times lower), acidification (100% lower) or eutrophication (12% lower). The results showed that films based on PP exert a substantial environmental impact compared to films based on chitosan, which are based on fossil fuels—the stage at which most pollutants are extracted from raw materials [138,139,140]. In most cases, carcinogens contribute to environmental pollution primarily due to the end-of-life stage, which mainly involves landfilling. In landfills, 94% of the gas produced is not collected for incineration or energy production, which causes the environmental burden associated with carcinogens. Moreover, short-term emissions from landfills should reach surface water, as well as groundwater pollutants [110]. Similarly, biopolymers are more environmentally destructive than petroleum-based plastics, particularly when it comes to eutrophication, ozone depletion, and non-carcinogenic health effects. PLA and TPS have been observed to be better at acidification, smog formation, eco-toxicity, and carcinogen. PLA’s eco-toxicity ranges from three times that of PP to 1.2 times that of PET. Further, PLA and TPS contribute more to eutrophication and ozone depletion than petroleum-based alternatives [110].

## 8. Present Research Obstacles and Future Prospects

It has been observed that there is currently a trend towards the consumption of bio-resourced, plant-based, and more natural, eco-friendly cosmetics, food, and beverages. To satisfy this trend, industries, as well as scientists and pharmacists, are focusing on the production, isolation, and application of natural products such as biopolymers as natural emulsifiers. Polysaccharide-based bio-surfactants, and other bio-based polymers are on this list, unlike their synthetic prototypes. Many such emulsifiers can form stable droplets of oil-in-water emulsions and are thus proven to be suitable for producing food and cosmetic products [141].

The main challenge in this field is the production cost of such biopolymers. The production and purification costs are still economically less favorable for the large-scale industrial application of biopolymers. Instrumentation and better methodologies are also needed. Funding for research devoted to biopolymers is a further obstacle [142].

However, we need to investigate and examine far more significant outcomes to successfully unveil several natural biopolymers as emulsifiers and emulsion stabilizers [15]. The natural biopolymers emulsifiers that possess better functionality, such as stability to freezing/thawing, protection of encapsulated components against chemical degradation, or controlled release properties, are very much effective in the food industry.

## 9. Conclusions

The dispersion of water in oil and vice versa is known as emulsion; emulsion with a globule diameter up to 20–200 nm is called nanoemulsion. The chemical substances that are added or used to stabilize nanoemulsions are called nanoemulsion stabilizers. In conclusion, a number of biopolymer stabilizers have been found to be more effective in various industries, such as food, pharmaceuticals, and cosmetics, than synthetic emulsion stabilizers. This is because the biopolymer emulsion stabilizer is less toxic; it is biodegradable, present in nature, cost-effective, and less immunogenic; and it offers nutritional benefits, and safety of use, handling, and transfer. Proteins and polysaccharides are frequently used biopolymers in emulsion synthesis. Biopolymers with lower concentrations and smaller sizes feature greater stabilizing properties and, generally, their stabilization of nanoemulsion is determined by temperature fluctuation, preservation time, pH value, ionic strength, processing technologies, etc. Additionally, biopolymers are also used to improve their nutritional value, increase self-life, and enhance texture and nutrient delivery in food products. Many plant proteins, such as pulse proteins (soybean proteins, pea proteins), cereal grain proteins, and whey proteins can be used in the synthesis of protein emulsion. Plant protein emulsion and animal protein emulsion are the best alternatives to synthetic nanoemulsion, so further research into plant protein emulsion will be more beneficial for biopolymer and emulsion stabilization. These highly applicable properties of biopolymers represent the strong prospects of biopolymers as emulsifiers in various advanced food and pharmaceutical technologies, as well as many other sectors such as biofuel production, biochemical fertilizer manufacturing, biodegradable plastics manufacturing, etc. Therefore, the use of biopolymer-based products will be very effective at maintaining the carbon cycle and will help to save the natural environment, as well as the whole Earth.

## Figures and Tables

**Figure 1 polymers-14-00127-f001:**
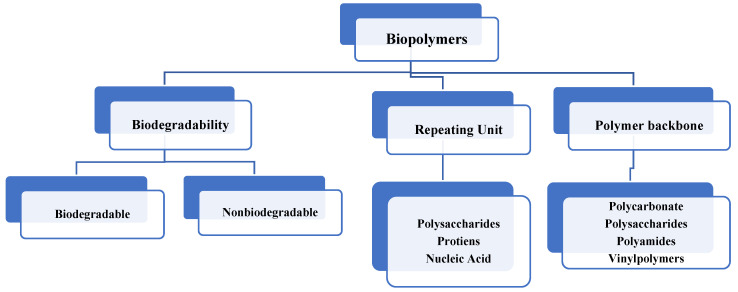
Overall classification of natural polymers [4].

**Figure 2 polymers-14-00127-f002:**
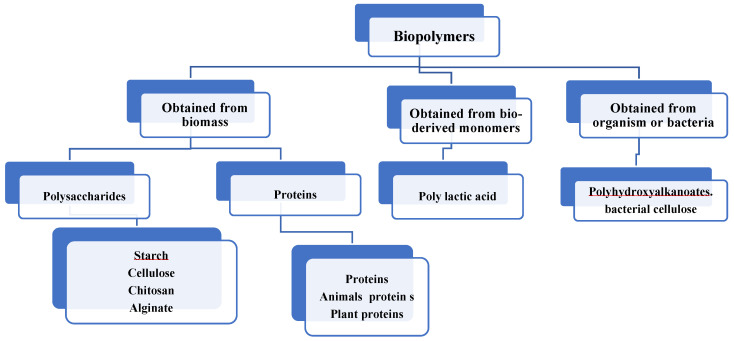
Classification of biopolymers based on sources [5].

**Figure 3 polymers-14-00127-f003:**
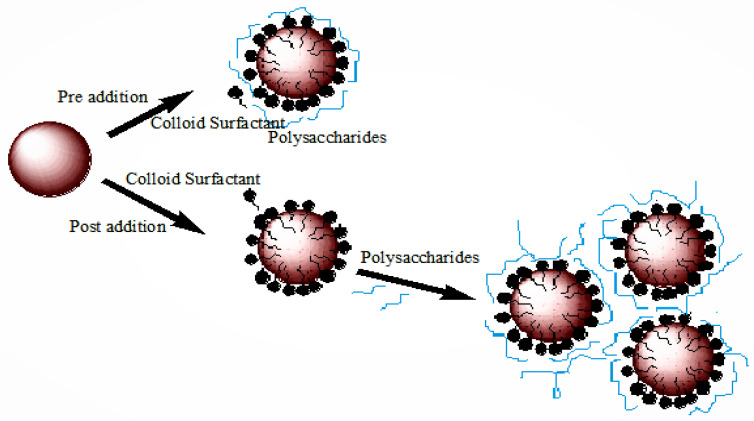
Stabilization of emulsion by using biopolymer stabilizer [5].

**Figure 4 polymers-14-00127-f004:**
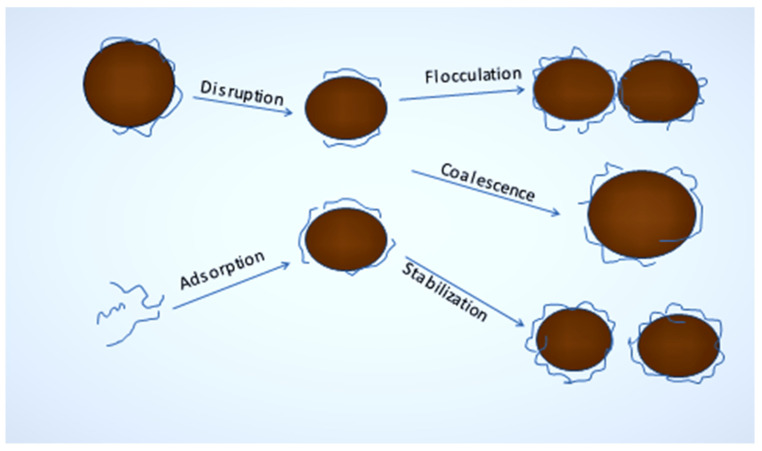
Overall views of instability emulsion [24].

**Table 1 polymers-14-00127-t001:** Biopolymers and biopolymer degradation [11].

S. N	Biopolymers	Degradation	Sources
1	Polysaccharides	Degradation by enzymes	Starch (wheat, potato, maize), Ligno-cellulosic products (wood, straw), Chitosan/Chitin.
2	Proteins and Lipids	Degradation by enzymes	Animals (casein, whey, gelation, collagen), Plants (zein, soy, etc.)
3	Polyhydroxyalkanoates (PHA)	Degradation by hydrolysis	Polyhydroxybutanoate (PHB), Polyhydroxy butyrate *co*-hydroxyvalerate (PHBV)
4	Polylactic Acid	Degradation by hydrolysis	Polylactic Acid
5	Petrochemical Polymers	Degradation by hydrolysis	Polycaprolactone (PCL), Polyester Amides (PEA)

**Table 2 polymers-14-00127-t002:** Differences between emulsion and nanoemulsion.

Properties	Emulsions	Nanoemulsions	References
Droplet size	Lager than nanoemulsions	20–200 nm	[16]
Stability	Thermodynamically unstable	Thermodynamically stable	[17]
Formation	By high shear homogenization methods	Micro-fluidization of emulsions	[18]
Viscosity	Higher viscosity than nanoemulsions	Lower viscosity than emulsions	[18]

**Table 3 polymers-14-00127-t003:** Comparison of nanoemulsion with droplet size.

Sources	Emulsification Techniques	Droplet Size	References
Fluids	Ultrasonic emulsification	24.21 ± 0.11 nm	[30]
Pastes	Emulsion inversion point method	<300 nm	[31]
Fogs	High-pressure homogenization	200–600 nm	[32]
Gels	Microfluidization	<100 nm	[33,34]
Fine liquid and solid particles in the air	Vertex mixing	282 nm	[35,36]
Topical	High-pressure homogenization	50–100 nm	[37]
Oral	Microfluidization	22 ± 4.0 nm	[38]
Intravenous	High-pressure homogenization	89.23 ± 7.2 nm	[39]
Intranasal, pulmonary, and ocular	High-pressure homogenization	8.4 ± 12.7 nm	[40]
Cosmetic industry	Ultrasonic emulsification	6–10 nm	[41]
Pesticide industry	Low-energy emulsification	~30 nm	[42]

**Table 4 polymers-14-00127-t004:** Comparative analysis of biopolymers and synthetic polymers.

Polymer	Type	Lifespan/Degradation Time	Mechanism of Degradation	Reference
Collagen types I, II, III	Bio/ Semi-synthetic	12 h	Enzymatic: collagenase	[124]
Cross-linked collagen	Semi-synthetic	>6 weeks	Enzymatic: collagenase	[125]
Alginate	Semi-synthetic	~80 days	Hydrolytic disintegration	[126]
Cross-linked chitosan	Semi-synthetic	>20 weeks	Enzymatic: chitosanase and lysosome	[127]
Hyaluronan films	Biopolymer	1 week to 4 months	Enzymatic: hyaluronidase	[128]
Braided silk	Biopolymer	6 weeks	Proteolysis	[129]
Polycaprolactone (PCL)	Synthetic	>24 months	Hydrolytic	[130]
PLA	Synthetic	>24 months	Hydrolytic	[130]
PHA/PHB	Synthetic	>24 months	Bacterial fermentation	[131]

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
