# Peer review of "A Review of Biopolymers’ Utility as Emulsion Stabilizers"

_polymers, 2021, doi:10.3390/polym14010127_

Round 1

Reviewer 1 Report

The aim of the review is to report the use of biopolymers as tension-active polymers for the preparation of emulsion.

The authors are not very cautious in the title because they claim their study to be systematic. The number of papers using biopolymers for the preparation of emulsions is so high that I doubt that this review could be systematic.

My own feeling is that I don’t like the word “biopolymers” because it is very confusing although this word is used everywhere in the state of the art. Does it mean bio-sourced, bio-degradable or bio-compatible? There is a trend to favor the meaning in favor of “bio-sourced polymer” but that can change from author to author. My understanding is that this is the meaning used by the authors of this review.

I was surprised to see the polyolefins in table 2 because they are typically oil-sourced polymers. I understand that they can be also bio-sourced in the case of polyethylene when the monomer used is bio-ethylene produced from bio-ethanol. Whatever, the authors mentioned these polymers for another reason akin to their biodegradability. A more critical discussion is needed because polyethylene is oxo-degradable in the presence of catalyst, which has been the topics of a lot of discussion in the past. They should not be included in the review or, at least, a mored detailed discussion on the criteria used to consider the polymers biodegradable is then needed.

Another reason why I don’t like the word biopolymer is that is too often used as keyword associated with a systematic good respect of the environment. The authors follow this trend as shown by the last sentence of their review: “the use of biopolymer-based products will be very effective to maintain the carbon cycle and it will help to save the natural environment and also the whole earth.  This is not scientifically rigorous because the impact on the environment must be assessed on the full life of the polymer and not just by one step (the production for bio-sourced polymer, the application step for biocompatible polymer or the end of life for biodegradable polymer). In other words, the impact on the environment must be assessed by considering the Life Cycle Analysis.

The use of biodegradable polymers for the improvement of the impact on the environment can be discussed in the review, which is not done in the paper. It is true that (bio)degradability avoids accumulating polymer waste in the environment for a very long time when they are dumped in nature. Nevertheless, after degradation, they are not valorized anymore in the frame of circular economy. This discussion must take into account the application. For a cosmetic application, when there is no possibility to collect the polymer after use, it is important that it is biodegradable. In a biomedical application, the biocompatibility criteria is more important because of the lower amount of material used compared to packaging and because the impact on heath is a first priority.

Finally, the word “bio-sourced” is also very confusing although if it is used a lot in the literature. The life of the carbon can be described as a cycle and, in a cycle, there is not starting point and thus no origin (no source). Oil is also produced from biomass in nature and is also bio-sourced! It is clearer to indicate the nature of the resource harvested in nature to obtain polymer (oil, agricultural resource, a forestry resource…) A clear distinction can be defined based on the renewability of the resource depending on the speed nature regenerates the resource compared with the speed at which humans harvest this resource in nature.

In the review, synthetic polymers are disregarded in the review. Nevertheless, synthetic polymers can sometimes be biodegradable, recyclable (when they can be collected and sorted) and they can have a better Life Cycle Assessment compared to bio-sourced polymers.

The paper reviews examples of applications based on a family of polymers used as surfactants but focus less on the principles of physico-chemistry used for the preparation of formulations (the emulsification process).

A general discussion discussing the advantages and limitations of the biopolymers reported in the review compared to synthetic polymers is missing. This kind of discussion is very important to make the review attractive for the reader and to avoid that the reviews become a listing of examples. Of course, some advantages and limitation are noted for specific examples, but an overview is missing.

My conclusion is that the scientific discussion is not enough critical and rigorous. The interest for the reader is limited due to the missing general discussion that I expect to find in a review to avoid that the review become a listing of examples. My own feeling is that the review is not mature for publication.

Author Response

First of all, we would like to thank the reviewers for the worthy comments. The comments indeed help us to improve the quality of the paper. We have tried to incorporate all the comments. All changes are noted in red. Please find below the reviewer comments and response.

Reviewer 1.

Reviewer Comments

Response

The aim of the review is to report the use of biopolymers as tension-active polymers for the preparation of emulsion.

 We are thankful to both the reviewers for reading, understanding and discussing different points of our review article for its inclusive scientific modification.

The authors are not very cautious in the title because they claim their study to be systematic. The number of papers using biopolymers for the preparation of emulsions is so high that I doubt that this review could be systematic.

With due respect to your kind comment, here we feel that we have written the review with full attention to existing relevant literature, care and dedication to the best of our abilities. So, we will be happy if the title remains unchanged. Although as per the reviewer’s suggestion we can change the title to " A  Review of Biopolymers' Utility as Emulsion Stabilizers"

My own feeling is that I don’t like the word “biopolymers” because it is very confusing although this word is used everywhere in the state of the art. Does it mean bio-sourced, bio-degradable or bio-compatible? There is a trend to favor the meaning in favor of “bio-sourced polymer” but that can change from author to author. My understanding is that this is the meaning used by the authors of this review.

We respect your personal feelings. Sir, there are still so many things questionable in science. With time modifications on even terminologies will be done. But for now, we must use those terms that are used by majority of people. There is really trend in society even in the domain of science to notify bio-resourced products as bio-products.  We used this term with same conventional thoughts. We have used the term with conventional meanings as they are commonly used.

I was surprised to see the polyolefins in table 2 because they are typically oil-sourced polymers. I understand that they can be also bio-sourced in the case of polyethylene when the monomer used is bio-ethylene produced from bio-ethanol. Whatever, the authors mentioned these polymers for another reason akin to their biodegradability. A more critical discussion is needed because polyethylene is oxo-degradable in the presence of catalyst, which has been the topics of a lot of discussion in the past. They should not be included in the review or, at least, a mored detailed discussion on the criteria used to consider the polymers biodegradable is then needed.

 It has been used there for their bio-degradability and as a consequence their less toxic impact towards nature. ‘Biopolymer’, the term itself is very much inclusive. It signifies the bio-based production of polymers as well as their bio-degradability. A more dedicated paper would rather be appropriate for this critical discussion. For this article it could exaggerate this article. We are thankful to the reviewer for his invaluable suggestion.

A whole new segment (7)’ Advantages and limitations of the biopolymers over synthetic polymers’ has been added with the manuscript where detail discussions are provided. We are hopeful that it will be helpful to all readers.

Another reason why I don’t like the word biopolymer is that is too often used as keyword associated with a systematic good respect of the environment. The authors follow this trend as shown by the last sentence of their review: “the use of biopolymer-based products will be very effective to maintain the carbon cycle and it will help to save the natural environment and also the whole earth.  This is not scientifically rigorous because the impact on the environment must be assessed on the full life of the polymer and not just by one step (the production for bio-sourced polymer, the application step for biocompatible polymer or the end of life for biodegradable polymer). In other words, the impact on the environment must be assessed by considering the Life Cycle Analysis.

The segment (7) Advantages and limitations of the biopolymers over synthetic polymers’ has been added with the manuscript where detail discussions are provided on comparative life cycle assessment of both synthetic and bio-polymers. We are thankful to the reviewer for his comment and making our article more scientifically enriched and helpful to readers of this article.

The use of biodegradable polymers for the improvement of the impact on the environment can be discussed in the review, which is not done in the paper. It is true that (bio)degradability avoids accumulating polymer waste in the environment for a very long time when they are dumped in nature. Nevertheless, after degradation, they are not valorized anymore in the frame of circular economy. This discussion must take into account the application. For a cosmetic application, when there is no possibility to collect the polymer after use, it is important that it is biodegradable. In a biomedical application, the biocompatibility criteria is more important because of the lower amount of material used compared to packaging and because the impact on heath is a first priority.

 There are so many aspects of a good grade ‘green chemical’. Bio-polymers are having many ecology and economy friendly properties. Notifying all the aspects and discussing those in a single review would make it excessive large and also difficult for us. Making this article concise and to the point was our primary target. We acknowledge and thank the reviewer for his insight for the improvement of our article. But we are sorry for not being able to add this discussion here.

Finally, the word “bio-sourced” is also very confusing although if it is used a lot in the literature. The life of the carbon can be described as a cycle and, in a cycle, there is not starting point and thus no origin (no source). Oil is also produced from biomass in nature and is also bio-sourced! It is clearer to indicate the nature of the resource harvested in nature to obtain polymer (oil, agricultural resource, a forestry resource…) A clear distinction can be defined based on the renewability of the resource depending on the speed nature regenerates the resource compared with the speed at which humans harvest this resource in nature.

We are thankful to the reviewer for reading, understanding and discussing points of our review article for its inclusive scientific modification.

In the review, synthetic polymers are disregarded in the review. Nevertheless, synthetic polymers can sometimes be biodegradable, recyclable (when they can be collected and sorted) and they can have a better Life Cycle Assessment compared to bio-sourced polymers.

The segment (7) Advantages and limitations of the biopolymers over synthetic polymers’ has been added with the manuscript where detail discussions are provided on comparative life cycle assessment of both synthetic and bio-polymers. We are thankful to the reviewer for his comment and making our article more scientifically enriched and helpful to readers of this article.

 The paper reviews examples of applications based on a family of polymers used as surfactants but focus less on the principles of physico-chemistry used for the preparation of formulations (the emulsification process).

A general discussion discussing the advantages and limitations of the biopolymers reported in the review compared to synthetic polymers is missing. This kind of discussion is very important to make the review attractive for the reader and to avoid that the reviews become a listing of examples. Of course, some advantages and limitation are noted for specific examples, but an overview is missing.

As per the reviewer’s good suggestion we have added this general discussion. The new segment (7)’ Advantages and limitations of the biopolymers over synthetic polymers’ has been added with the manuscript where detail discussions are provided. We are again showing gratitude towards the reviewer for his comment which helped us understanding the need for improvising our article.

My conclusion is that the scientific discussion is not enough critical and rigorous. The interest for the reader is limited due to the missing general discussion that I expect to find in a review to avoid that the review become a listing of examples. My own feeling is that the review is not mature for publication.

We have tried to answer all the quires of the reviewer. We also have done changes that were suitable for the betterment and scientific upgradation of the article. We hope the reviewer will be happy with our explanations and modifications and will suggest in favor of us for publishing the article. Once again, we thank the reviewer of giving time and understanding and our article and giving suggestions for its modification.

Reviewer 2 Report

Despite the fact that there have been numerous publications in the field of Biopolymers Utility as Emulsion Stabilizers, the authors took a chance and wrote an impressive manuscript. However, before it can be considered for publication, it must undergo further revision according to the following comments.

  1. To support the claims statements included in the first paragraph of the introduction, certain references are required.
  2. In Table 1, sources column - Remove the bullet points and rewrite it as a sentence, for instance. For example, Starch (Wheat, Potato and Maize), Ligno-cellulosic products (Wood and Straw), Proteins and Chitosan/Chitin.
  3. Figure 2 is not required, because the same information is repeated in the main text. Remove the bullet points and replace them with connecting sentences in a single paragraph.
  4. To support the Introduction section, authors should refer to a few recently published articles and cite those sources. Because the majority of the references in the introduction section are older than five years.
  5. Table 2, droplets size to be changed as Droplets size  
  6. Figures 3 and 4 are unclear; they appear to have been copied from a published article. If possible, the author should use software to create their own figure. Otherwise, get rid of it. Figures 1A and 1B should also be software-modified to make them more appealing.
  7. Synthesis of polysaccharides emulsion – This section requires references.
  8. Glycosyl with polysaccharides – Additional references are required to support the information in this section.
  9. Percent or %, maintain a consistent formatting style throughout the manuscript.
  10. Include a section on current research obstacles/challenges and potential future directions on biopolymer-based products.
  11. In addition, a thorough grammar check should be performed throughout the manuscript.

Author Response

First of all, we would like to thank the reviewers for the worthy comments. The comments indeed help us to improve the quality of the paper. We have tried to incorporate all the comments. All changes are noted in red. Please find below the reviewer comments and response.

Reviewer 2.

Reviewer Comments

Responses

  1. To support the claims statements included in the first paragraph of the introduction, certain references are required.

Sir, we have added 2 new references.

  1. In Table 1, sources column - Remove the bullet points and rewrite it as a sentence, for instance. For example, Starch (Wheat, Potato and Maize), Ligno-cellulosic products (Wood and Straw), Proteins and Chitosan/Chitin.

We are thankful to the reviewer for worthy comments. This has indeed helped to improve the quality of the paper.

In table 1, source column the bullet point has been removed as a given example, please kindly visit in page.3. Table 1.

  1. Figure 2 is not required, because the same information is repeated in the main text. Remove the bullet points and replace them with connecting sentences in a single paragraph.

Figure 2 has been replaced and all bullet point has been replaced with sentence.

  1. To support the Introduction section, authors should refer to a few recently published articles and cite those sources. Because the majority of the references in the introduction section are older than five years.

We have added many new current references.

  1. Table 2, droplets size to be changed as Droplets size  

We would like to thank the reviewer for highlighting this word. This has indeed helped to improve the quality of the paper.

'droplets' has been changed into 'Droplets' size in table 2.

  1. Figures 3 and 4 are unclear; they appear to have been copied from a published article. If possible, the author should use software to create their own figure. Otherwise, get rid of it. Figures 1A and 1B should also be software-modified to make them more appealing.

Figure 3 and 4 have been modified. Figure 1A and 1B has also been modified.

  1. Synthesis of polysaccharides emulsion – This section requires references.

References are added here in this segment.

  1. Glycosyl with polysaccharides – Additional references are required to support the information in this section.

References are added here in this segment.

  1. Percent or %, maintain a consistent formatting style throughout the manuscript.

The percent has been changed into %, please kindly visit in section 4.3.

  1. Include a section on current research obstacles/challenges and potential future directions on biopolymer-based products.

A new segment has been written as per the reviewer’s suggestion.

  1. In addition, a thorough grammar check should be performed throughout the manuscript.

We have gone through the paper several times to modify the mistakes to the best of our knowledge.

Round 2

Reviewer 1 Report

The authors submitted a revised version of their paper with a major modification, by adding a new section in the paper. The answers of the authors are convincing, and they considered my remarks in the updated version. I can follow their claims on the environmental impact added in the revised version.

Reviewer 2 Report

The authors carefully considered the comments from the reviewers and made changes in the manuscript. Hence, I recommend that the revised manuscript may be accepted for publication in its current form.